# Evaluation of Single-Trial Classification to Control a Visual ERP-BCI under a Situation Awareness Scenario

**DOI:** 10.3390/brainsci13060886

**Published:** 2023-05-31

**Authors:** Álvaro Fernández-Rodríguez, Ricardo Ron-Angevin, Francisco Velasco-Álvarez, Jaime Diaz-Pineda, Théodore Letouzé, Jean-Marc André

**Affiliations:** 1Departamento de Tecnología Electrónica, Instituto Universitario de Investigación en Telecomunicación de la Universidad de Málaga (TELMA), Universidad de Málaga, 29071 Malaga, Spain; afernandezrguez@uma.es (Á.F.-R.); fvelasco@uma.es (F.V.-Á.); 2Thales AVS Bordeaux, 33700 Mérignac, France; jaime.diazpineda@fr.thalesgroup.com; 3Laboratoire IMS, CNRS UMR 5218, Cognitive Team, Bordeaux INP-ENSC, 33400 Talence, France; tletouze@ensc.fr (T.L.); jean-marc.andre@ensc.fr (J.-M.A.)

**Keywords:** brain–computer interface (BCI), electroencephalography (EEG), event-related potential (ERP), air traffic controller, situation awareness

## Abstract

An event-related potential (ERP)-based brain–computer interface (BCI) can be used to monitor a user’s cognitive state during a surveillance task in a situational awareness context. The present study explores the use of an ERP-BCI for detecting new planes in an air traffic controller (ATC). Two experiments were conducted to evaluate the impact of different visual factors on target detection. Experiment 1 validated the type of stimulus used and the effect of not knowing its appearance location in an ERP-BCI scenario. Experiment 2 evaluated the effect of the size of the target stimulus appearance area and the stimulus salience in an ATC scenario. The main results demonstrate that the size of the plane appearance area had a negative impact on the detection performance and on the amplitude of the P300 component. Future studies should address this issue to improve the performance of an ATC in stimulus detection using an ERP-BCI.

## 1. Introduction

Brain–computer interfaces (BCIs) are a type of technology that employs brain activity to establish a communication channel between a user and a device [1]. This communication channel can be used either for the user to control different devices through a brain signal (e.g., a wheelchair or a virtual keyboard), or for monitoring the user’s cognitive state (e.g., stress or mental workload) [2]. The most common methodology employed by a BCI to obtain a user’s brain activity is electroencephalography (EEG) [3]. Some of the advantages of EEG are its adequate temporal resolution, relatively low cost and non-invasiveness [4]. BCIs have been used in several areas such as clinical or leisure applications [5]. However, some research has shown how BCIs can also be used effectively to assist decision making or to monitor the state of a user during a surveillance task in a situational awareness (SA) context [6,7]. This is the focus of the present research. SA involves the interpretation of environmental factors and occurrences in relation to time and location, as well as the prediction of their future states. Specifically, according to [8], SA can be approached through a hierarchical framework consisting of three levels: (i) perception of elements in the current situation; (ii) understanding of the current situation; and (iii) projection of a future situation. Today, eye-tracking is used to determine whether a user has perceived a specific stimulus on the screen [9,10]. However, this approach cannot guarantee that the user has initiated a cognitive process of detecting and understanding the stimulus. For this reason, in the present work, the use of a BCI system is proposed.

An air traffic controller (ATC) is a scenario in which a professional operator directs planes on the ground and through a given section of controlled airspace. ATCs’ main goals are to avoid collisions, streamline and arrange aviation traffic and give pilots information and assistance. Thus, an ATC could be an appropriate SA scenario for the use of BCIs to assist decision making, in which a user must be aware of different cues and react accordingly [11,12,13]. This paper focuses on BCI applications for an ATC. The aim of a BCI for an ATC should be to improve the safety and accuracy of the system that is being controlled. In general, two types of BCI systems can be distinguished to meet these objectives: passive and active. A passive BCI aims to recognize the state of the user during task execution, so that the system can recognize when the user is, for example, tired or has a high mental workload [13]. It would be valuable for the system to recognize these user cognitive states, as they could be indicative of future errors in detecting critical cues for the prevention of potential incidents [14,15]. In the ATC context, an active BCI would be intended to assist with decision making (e.g., to know if the user has perceived the appearance of a new relevant element, such as a warning message). To our knowledge, there is no work that has employed an active BCI for the detection of new elements in the ATC scenario. Therefore, the present work focuses on active BCIs and the first level of the SA framework, i.e., the perception of elements in the current situation. Specifically, the ability to detect the appearance of new key elements—for example, new planes on the map—through the user’s EEG signal that controls the system is the focus of this work. This could be very useful in a hypothetical danger scenario in which the BCI can assess the attention capabilities of the controller.

The visual stimuli to be attended to by ATCs are planes in a virtual representation of a map, so the present work uses visual event-related potentials (ERPs) recorded through EEG as the detection input signal for the BCI. Visual ERPs are potential changes in the electrical activity of the brain elicited by the presentation of visual stimuli. Hence, the objective of an ERP-BCI is to detect the desired or attended stimulus based on the user’s brain signals. The main component used by these systems is the P300. This is a positive deflection in the amplitude of the brain’s electrical signal that begins approximately 300–600 ms after the presentation of a stimulus that the user is expecting (target) [16]. However, an ERP-BCI generally uses all possible ERPs involved in the observed time interval (e.g., P2, N2 or N400). These ERPs can be influenced by the properties of the stimuli that elicit them, such as the type [17], size [18,19] or luminosity [20]. These previous findings should be considered when designing a visual ERP-BCI for an ATC. There are several differences between applications commonly controlled by a visual ERP-BCI—such as wheelchairs [21] or virtual keyboards [22]—and an ATC. Two of the most relevant differences are (i) the number of presentations of the target stimulus and (ii) the location of the appearance of the target stimulus. On the one hand, in most visual ERP-BCI applications, target stimuli are presented several times to maximize the probability that they are correctly selected. However, for an ATC—and, in general, in any application in which alert messages are presented—it is important that the target stimulus can be recognized after a single presentation. In other words, the visual ERP-BCI operates with single-trial classification, which refers to when the detection of a target stimulus is selected through a single presentation of the stimulus (e.g., a particular letter in a writing system). This is a challenge because ERP-BCIs usually need several presentations of the stimulus to obtain satisfactory performance. Several presentations of the stimulus are necessary to correctly discern the specific components of the EEG signal from the noise (e.g., muscle artifacts). As more presentations are made, the noise level decreases and, therefore, the ERP components linked to the presentation of a target stimulus are better observed. Nevertheless, some previous ERP-BCI proposals focused on the use of a single trial have shown adequate performance (~80% accuracy [23,24,25]). However, to our knowledge, these authors did not present the characteristics that could hinder the performance of an ATC (e.g., the use of a stimulus-rich map as the background, moving planes or target stimuli of a reduced size, such as the planes to be perceived). Hence, it might be interesting to explore the use of single-trial classification in the context of an ATC. On the other hand, in most visual ERP-BCIs, the target stimuli are usually presented in a specific location previously known to the user, but new planes may appear in an unknown location for an ATC. Thus, it would be interesting to study whether the size of the appearance surface has an influence on performance.

The present work aimed to evaluate the effect of different visual variables on the performance of a visual ERP-BCI in an SA scenario to detect the appearance of new planes by an ATC. Because the use of an active BCI to assist an ATC is a novel approach, two experiments were performed to explore this topic. Experiment 1 was an initial approach to test single-trial classification and a BCI single-character paradigm (SCP), in which the stimuli were individually presented one after the other at different locations on the screen [26]. Experiment 1 served to determine the effect of (i) presenting two types of stimuli (faces versus radar planes) and (ii) knowing (or not knowing) the specific location where the target stimulus would appear. Experiment 2 used an ATC environment to test the effect on the performance of detecting new planes with (i) the use of a different (or similar) color from those of the rest of the planes that were already moving in the interface and (ii) the size of the area to be watched by the user.

## 2. Materials and Methods

### 2.1. Participants

Each experiment involved ten participants (Experiment 1, 22.33 ± 1.87 years old, three women, named E101–E110; Experiment 2, 24.44 ± 2.01 years old, four women, named E201–E210). The experience of the participants in the control of the ERP-BCI was variable. Of note, none of the participants had experience with using an ATC. All subjects gave their written informed consent on the anonymous use of their EEG data. They declared having normal or corrected-to-normal vision. This study was approved by the Ethics Committee of the University of Malaga and met the ethical standards of the Declaration of Helsinki.

### 2.2. Data Acquisition and Signal Processing

Signals were recorded through eight active electrodes, namely Fz, Cz, Pz, Oz, P3, P4, PO7 and PO8 (10/10 international system). A reference electrode was placed on the left mastoid, and a ground electrode was placed at AFz. An actiCHamp amplifier (Brain Products GmbH, Gilching, Germany) was used, with a sample rate of 250 Hz. The data were collected with BCI2000 [27]. To reduce the impact of EEG noise, a band pass filter was applied to the EEG signal between 0.1 and 30 Hz using a first-order infinite impulse response filter for the high pass and a second-order Butterworth IIR filter for the low pass. Additionally, a notch filter was set at 50 Hz using two third-order Chebyshev filters.

The visual ERP signal was used for controlling the BCI. Although no user training is required for visual ERP paradigms, calibration is necessary to determine subject-specific parameters for the experimental task. To accomplish this, each subject’s EEG was analyzed using stepwise linear discriminant analysis (SWLDA) with the BCI2000 tool named P300Classifier [28]. The P300 component is the most readily discernible ERP in these types of systems, hence the name of the tool. The SWLDA algorithm determines only statistically significant variables for the final regression, utilizing multiple linear regressions and iterative statistical procedures. The default configuration had a maximum of 60 features, and 0.1 and 0.15 were used as the maximum *p*-values for the respective inclusion and exclusion of a feature in the model. The analyzed time interval (i.e., the epoch length) was set at the default value of 0–800 ms after stimulus presentation. Consequently, subject-specific weights for the classifier were obtained and applied to the EEG to determine the target stimulus that the subjects attended to. It is worth noting that, recently, other decoding systems based on machine learning, such as deep learning, have been employed in BCI systems, showing promising results (refer to the review in [3] for details). However, the objective of this study does not pertain to signal processing algorithms but to conducting an initial test utilizing standard BCI software to control an ATC. Hence, the standardized SWLDA from BCI2000 was utilized as the classifier.

### 2.3. Experimental Conditions

The study of plane detection in an ATC scenario is a novel issue; to the best of our knowledge, there have been no studies on it. Therefore, a gradual, successive approach was applied in this study. Because the study was progressive, the protocols varied and were tailored as the research progressed. Both experiments employed the same hardware. An HP Envy 15-j100 laptop was used (2.20 GHz, 16 GB, Windows 10), but the display was an Acer P224W screen of 46.47 × 31.08 cm (16:10 ratio), connected through HDMI, at a resolution of 1680 × 1050 pixels. The refresh rate of the screen was 60.014 Hz. The distance between the user’s point of view and the screen was ~60 cm.

#### 2.3.1. Experiment 1

Before controlling the ATC using the user’s EEG signal, the effects of the type of used stimulus and whether the participant knew where the stimulus would appear were determined. If an acceptable performance was not found in these simpler conditions, it would be difficult to find good performance in the ATC scenario. The conditions used for Experiment 1 are described below.

The interface was displayed using the BCI2000 (3.6 R5711.1) software [27]. The paradigm used for the stimulus presentation was based on the SCP [26] and single-trial classification. This paradigm represents a first step to validate the use of an active BCI for the detection of a stimulus presented only once at a specific position on the screen (as in the case of plane detection for an ATC). Under the SCP, each stimulus is presented serially at a different location on the display. Nine stimuli were used, so there were also nine possible locations (one per stimulus), arranged in a 3 × 3 matrix. The employed visual stimuli differed according to the experimental condition, but they were all 3.4 × 3.4 cm (3.25° × 3.25° at 60 cm) (Figure 1). Target and non-target stimuli were serially displayed on a black background.

The following experimental conditions were used:E1-faces. In this condition, the used stimuli were red celebrity faces with a white square background, a type of stimulus that was suggested by recent work as one of the most appropriate to obtain high accuracy in the control of a visual ERP-BCI [29]. Both target and non-target stimuli were presented, and the user knew in advance the exact position of the appearance of the target stimulus.E1-planes. This condition was the same as E1-faces (the presence of target and non-target stimuli, and the user knew the specific location of the target stimulus) but employed symbols similar to those used for planes on radars.E1-known. In this condition, the stimuli were also radar planes, and the user knew in advance the exact position of the target stimulus. However, the non-target stimuli were not presented.E1-unknown. This condition was similar to E1-known, as it also employed radar plans, and non-target stimuli were not presented. However, in this condition, the user did not know in advance where the target stimulus would appear.

The aim of these conditions was to study the effect of two factors on system performance when detecting the presence of specific target stimuli in the interface based on the user’s EEG signal. On the one hand, the comparison between E1-faces and E1-planes allowed evaluating the effect of the type of stimulus. On the other hand, the comparison between E1-known and E1-unknown allowed evaluating the effect of knowing in advance the exact location of the appearance of the target stimulus.

#### 2.3.2. Experiment 2

This experiment was designed after evaluating the results of Experiment 1, which, in summary, validated that the system was able to detect the appearance of new target stimuli at the interface with a single presentation (single-trial classification) and without prior knowledge of the location where it would appear. Therefore, the goal of Experiment 2 was to transfer the results into an ATC scenario to study the ability of the system to detect the presentation of new target stimuli (i.e., planes).

Due to the limitations of BCI2000 to modify the display as required in this experiment, the Processing software was used to simulate an ATC scenario [30]. Processing is a graphic software coded in Java that was synchronized in time with BCI2000 through a UDP port (using the BCI2000 “watches” tool) and received the temporal instant in which the target stimulus was presented in BCI2000. The paradigm consisted of presenting a background video (extracted from https://www.flightradar24.com, accessed on 10 May 2022) over which the target stimulus to be attended by the user appeared. The background video showed an ATC map that covered the full screen, and different yellow planes could be seen. The site chosen for the video recording was a 129 × 80 km area containing two Paris airports (Charles de Gaulle and Paris-Orly) around 11:00–12:00 local time, so it could be considered an area with high plane concurrence (Figure 2). The target stimulus that the user had to attend to was the appearance of new planes at a random location. The complete area where the planes could appear was 42.93 × 27.74 cm (i.e., the entire screen excluding 1.67 cm horizontal and 1.77 cm vertical margins). Subsequently, this inner rectangle was divided into nine cells, forming a 3 × 3 matrix, of 14.31 × 9.25 cm each. Depending on the experimental condition, the user was told (or not told) in which cell the next target plane would appear; within this cell, the position of the plane would be random and unknown to the user. After a target plane appeared, it moved in a random but constant direction and speed, simulating the planes already present in the video. To be sure that the planes remained on the screen for the required amount of time, the planes appearing in some of the border cells were moving in a random direction but opposite to the screen margins. The size of the planes to be attended to by the participant was 1 × 1 cm (0.95° × 0.95° at 60 cm). Specifically, three conditions were tested in this experiment:E2-RS. In this condition, the plane was a different color (red) from those of the planes that were previously on the screen (yellow), and it appeared in a relatively small indicated area (one of the nine possible cells mentioned above). To indicate in which specific cell the plane would appear, a semi-transparent red rectangle was displayed over that cell (14.31 × 9.25 cm) for 1 s. The plane was displayed in red and faded to yellow within 2 s. The reason for this color change was that, when the new target plane appeared, it would be the only one in red on the screen, and the previous plane in red would then be the same color as the others (i.e., yellow).E2-YS. This condition was similar to E2-RS; the only difference was that the target planes were presented using the same color of the planes that were already on the screen (i.e., yellow).E2-RL. This condition was similar to E2-RS; the only difference was that the semi-transparent red rectangle indicating the appearance of the airplanes occupied the complete area corresponding to the nine cells (42.93 × 27.74 cm). Therefore, the user did not know in which specific cell the new plane would appear.

**Figure 2 brainsci-13-00886-f002:**
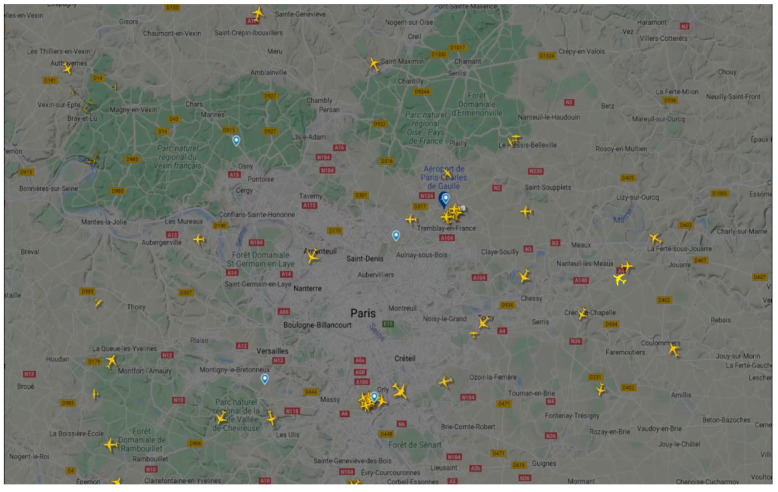
Screenshot of the video used in Experiment 2 for the air traffic controller (ATC) on which the new target planes to be attended to by the user appeared. The ATC video was extracted from https://www.flightradar24.com, accessed on 10 May 2022.

These three conditions allowed evaluating the effect of two factors on plane detection by analyzing the EEG signal of the user controlling the system: (i) the selection of a target plane of a similar or different color to those of the other planes already present by the ATC (E2-RS versus E2-YS), and (ii) the size of the area to be surveilled in which the target plane would appear (E2-RS versus E2-RL).

### 2.4. Procedure

Experiment 1 and Experiment 2 started in the same way: The participant arrived at the laboratory, the session with the tests to be performed was explained, he/she signed the informed consent form, the EEG electrodes and cap were placed, and the tasks could begin. Likewise, both experiments used an intra-subject design, so all users of each experiment went through all the conditions of that experiment. Each condition consisted of two exercises: (i) a calibration task to adapt the system to the user and (ii) an online task in which the system intended to detect the appearance of target stimuli. The main difference between the calibration and online tasks was that, in the online task, the user had feedback on his/her performance (i.e., whether the target stimulus had been correctly detected by the system) because their specific parameters (i.e., the weights for the P300Classifier) were already calculated after the calibration task. The terms used to detail the procedure of the experiments included the following. A run is the process to detect a single target stimulus. To complete a run, all the stimuli that compose the interface must be presented. A block is the interval from when the interface is started until it stops automatically; it is composed of the different runs made by the user.

#### 2.4.1. Experiment 1

Experiment 1 was divided into two consecutive sessions: a first session with conditions E1-faces and E1-planes, and a second session with conditions E1-known and E1-unknown. The order of the conditions of each session was counterbalanced among the subjects. The approximate duration of Experiment 1 was 80 min from the time the participant arrived at the laboratory until the end of the tasks. The four conditions used in this experiment had similar timing. Before the start of each block, there was a waiting time of 1920 ms, after which the different runs began. Moreover, at the beginning of each run (except for E1-unknown), a message was presented in Spanish (“Atiende a:” [Focus on:]) for 960 ms, after which the stimulus to be attended to was presented for another 960 ms. For E1-unknown, this information was replaced by a black background for 1920 ms. Before the first stimulus of the run was presented, all conditions included a pause time of 1920 ms. The stimulus duration was 384 ms, and the inter-stimulus interval (ISI) was 96 ms, resulting in a stimulus onset asynchrony (SOA) of 480 ms. Likewise, in the online task in all conditions, a message was presented at the end of each run (“Resultado:” [Result:]) for 960 ms, after which the stimulus selected by the system was presented for 960 ms. The attention and result messages were accompanied by an auditory cue to facilitate the user’s attention to the task. For both the calibration and online tasks, a pause time of 1920 ms was added. The specific procedure for the E1-faces and E1-planes conditions was identical, as was the specific procedure for E1-known and E1-unknown, so the particularities of each condition in this experiment are detailed below.E1-faces and E1-planes. The calibration task consisted of three blocks of six runs of 55 s each (Figure 3). In each block, the following stimuli were selected from left to right: for the first block, the three stimuli in rows 1 and 2; for the second block, the stimuli in rows 2 and 3; and for the third block, the stimuli in rows 1 and 3. Each block of the calibration task had a duration of 55 s. The online task consisted of presenting as target stimuli all stimuli of the interface in row-major order, i.e., nine runs in one block, which had a duration of 111 s (E101 and E102 performed 18 runs instead of 9).E1-known and E1-unknown. The calibration task consisted of 16 blocks of one run, resulting in a duration of 11 s per block (Figure 4). The online task used five blocks of one selection, with a duration of 14 s per block (E101 and E102 performed 10 blocks of one selection). For both tasks, the target stimulus order to be attended to was randomly selected with replacement.


**Figure 3 brainsci-13-00886-f003:**
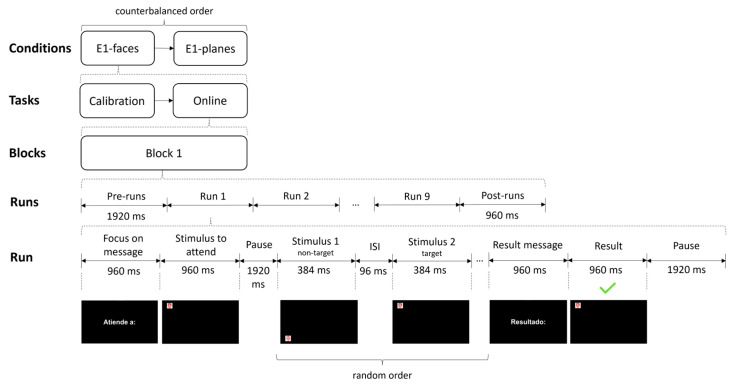
Procedure and timing used in the E1-faces and E1-planes conditions of Experiment 1. Specifically, the figure shows the execution of the first run of the E1-faces condition during the online task. ISI stands for inter-stimulus interval.

#### 2.4.2. Experiment 2

During this experiment, each participant tested the three previously detailed conditions in one session: E2-RS, E2-RL and E2-YS (Figure 5). The order of the conditions was counterbalanced and equally distributed among the subjects. The approximate duration of Experiment 2 was 60 min from the time the participant arrived at the laboratory until the end of the tasks. The three conditions used in this experiment had similar timing. Before the start of each block, there was a waiting time of 5000 ms, after which the different runs began. In addition, at the beginning of each run, either in the calibration or online task, a semi-transparent red rectangle was presented (1000 ms), indicating the appearance area of the target plane (restricted to one of the nine possible cells in E2-RS and E2-YS, or the combination of all of them in E2-RL). After the appearance of the red rectangle, there was a 2849 ms pause, after which there was a 4032 ms period during which the new plane could appear. After this period, there was a post-run pause of 5000 ms. In the online task, within this post-run pause, the user was given feedback on whether the system had correctly detected the appearance of the new plane. This feedback was indicated by an image, presented for 3000 ms, of a green thumbs-up if the system had detected the new plane or a red thumbs-down if the system had not detected the new plane. Both the semi-transparent red rectangle and the feedback images were accompanied by an audio cue to facilitate the participant’s attention to the task. For the background videos simulating the ATC, three distinct 4 min-long videos were used, one per condition and always in the same order. The same video was used for each block of the same condition. However, because the conditions were counterbalanced, the same video was not always associated with the same condition for all participants. For example, participant P01 could use video 1 with E2-RS, video 2 with E2-YS and video 3 with E2-RL, and for P02, this order could be video 1 with E2-RL, video 2 with E2-RS and video 3 with E2-YS.

The calibration task consisted of 32 runs in 4 blocks of 8 runs; each block had a duration of 108 s. The online task consisted of 20 runs in one block; this block had a duration of 263 s. For each run, the cell where the next target plane would be displayed was randomly chosen from the possible nine available cells. The exact location inside the cell where the plane would appear was also randomized.

### 2.5. Evaluation

Several variables were analyzed to evaluate the effect of the different factors manipulated in the experiments. In Experiment 1, the accuracy and amplitude of the ERP waveform were used. In Experiment 2, accuracy and ERP waveform were used, as well as the number of target planes missed (i.e., non-perceived) by the user.

#### 2.5.1. Accuracy

In all conditions, the classifier had to select a target stimulus from nine possible stimuli (including E1-known, E1-unknown and the three conditions of Experiment 2, in which the non-target stimuli were invisible to the user). The accuracy (%) corresponds to the percentage of correct selections divided by the total number of selections made. The accuracy was calculated for the online task of each condition. For Experiment 1, a paired sample *t*-test was used to compare the conditions. For Experiment 2, a repeated measures ANOVA was used to evaluate whether there were significant differences between conditions. If there were, the Bonferroni correction method for multiple comparisons was applied.

#### 2.5.2. Amplitude (µV) of the ERP Waveform

All analyses reported below regarding the ERP signal were carried out using EEGLAB software [31]. Once all the data had been registered and the session had ended, artifacts in the data were corrected through the artifact subspace reconstruction (ASR) algorithm using the default settings in EEGLAB and the Riemannian distance [32]. The amplitude of the target stimulus signals (μV) in the calibration task was evaluated to observe how the ERPs related to that apparition of the target stimulus were affected by the application of different experimental conditions. A time interval of −200 to 1000 ms was evaluated, using −200 to 0 ms as a baseline. Permutation-based statistics (non-parametric) were used to compare the amplitude (μV) between both types of stimuli obtained in all channels for each paradigm [31]. These analyses were also corrected using the false discovery rate (FDR) method [33], because multiple channels and intervals were compared simultaneously.

#### 2.5.3. Target Planes Missed

In Experiment 2, at the end of each block—both in the calibration task and in the online task—the participant was asked to write on a form the number of target plane apparitions he/she had perceived. Because the number of target stimuli presented was known, it was also possible to determine the number of missed planes, which indicates the user’s ability to perceive the target stimuli in each condition. Only eight participants (E203–E210) completed this measure, as it was implemented after two participants had completed the experiment (E201 and E202). Friedman’s test, a non-parametric test for the comparison of three or more related samples, was used to evaluate whether there were significant differences between conditions. If there were, then Conover’s post hoc test, with the Bonferroni correction method for multiple comparisons, was applied.

## 3. Results

### 3.1. Experiment 1

In this experiment, two factors were evaluated: (i) the stimulus type (faces versus radar planes), using visible non-targets; and (ii) the knowledge of the location of the stimulus to attend to before it appears (known versus unknown), using the radar plane stimulus type and invisible non-target stimuli.

#### 3.1.1. Accuracy

First, to test the effect of the type of stimulus on accuracy, the E1-faces and E1-planes conditions were compared (76.67 ± 30.75% and 73.89 ± 25.67%, respectively). The paired samples *t*-test showed that there was no significant difference between the conditions (*t* (9) = 0.409; *p* = 0.692). Therefore, it seems that the type of stimulus does not have a significant impact on performance. Second, to test the effect of prior knowledge of the stimulus location, the E1-known and E1-unknown accuracies were compared (79 ± 25.1% and 76 ± 15.06%, respectively). The paired samples *t*-test showed that knowing the location of the stimulus beforehand did not significantly affect accuracy (*t* (9) = 0.635; *p* = 0.541). Therefore, this experiment shows that knowing where to attend to the incoming target did not affect performance.

#### 3.1.2. Amplitude (µV) of the ERP Waveform

Regarding the E1-faces and E1-planes conditions, the registered signal does not allow for a clear observation of the ERPs commonly expected in this type of oddball paradigm-based study. However, considering the use of a single trial, it is normal for the signal to have more noise than initially expected. The only component that can be clearly differentiated is P300 for channels Pz, Oz, P3, P4, PO7 and PO8 with a peak between 400 and 600 ms after the presentation of the target stimulus (Figure 6A).

There were no significant differences between the target stimulus amplitude of both conditions. Therefore, the type of stimulus employed by the conditions did not show to significantly influence the target stimulus amplitude waveform. Regarding the E1-known and E1-unknown conditions, although it may be speculative to discriminate any particular component, it is possible that P300 can be observed around 350–600 ms for channels Pz, Oz, P3, P4, P7 and PO8. Again, the analyses indicated non-significant differences in relation to the target amplitude between conditions. Hence, it cannot be stated that the lack of knowledge regarding the location of the target stimulus affects any of the involved components.

### 3.2. Experiment 2

In this experiment, two factors were studied: (i) the effect of the color of the stimulus that appeared on the ATC (in red, different from the rest of the planes of the ATC, or in yellow, similar to the rest of the planes already present), and (ii) the effect of the size of the appearance surface of the target planes.

#### 3.2.1. Accuracy

The accuracy obtained for each condition was as follows: E2-RS, 64.5 ± 24.77%; E2-YS, 67.5 ± 10.69%; and E2-RL, 41 ± 20.25%. The repeated measures ANOVA indicated that there were significant differences between the conditions evaluated in reference to accuracy (*F* (2) = 9.368, *p* = 0.002). Specifically, these differences were found between E2-RS and E2-RL (*p* = 0.024) and between E2-YS and E2-RL (*p* = 0.011), but not between E2-RS and E2-YS (*p* = 1). Therefore, the results related to accuracy proved that the size of the display area has a negative influence on performance to detect the appearance of new planes using an ERP-BCI. However, despite what was expected, it seems that, in a relatively small area, the color of the plane had no significant effect on the accuracy of the system.

#### 3.2.2. Amplitude (µV) of the ERP Waveform

As shown in Figure 7, the three tested conditions exhibited a positive potential in all channels, with a peak at ~520 ms, which was likely produced by the P300 component. The analyses showed significant differences in amplitude differences between conditions. Specifically, the pattern appears to be consistent across all channels; the amplitude of the target stimulus related to the E2-Ys condition seems to obtain the highest levels, closely followed by E2-RS, with E2-RL in last place with noticeably lower levels of amplitude. Therefore, these results indicate that the P300 amplitude is primarily influenced by the size of the surface to be surveyed, rather than by the salience of the stimulus (manipulated through the color of the plane).

#### 3.2.3. Target Planes Missed

The number of target planes missed for each condition was as follows: E2-RS, 0.25 ± 0.46; E2-YS, 0.25 ± 0.71; and E2-RL, 2.38 ± 2.45. The Friedman’s test indicated significant differences between the conditions (*χ*^2^(2) = 4.625, *p* = 0.013). Specifically, multiple comparisons indicated that condition E2-YS showed a significantly lower number of errors compared to E2-RL (*p* = 0.048), but not between E2-RS and E2-YS (*p* = 1), nor between E2-RS and E2-RL (*p* = 0.084). However, it should be noted that only one participant lost two airplanes in condition E2-YS, whereas in E2-RS, two participants lost one airplane each, and in E2-RL, all participants except one lost at least one airplane. Therefore, it appears that the size of the area to be monitored indeed affects the ability to detect airplanes, as was the case with accuracy.

## 4. Discussion

### 4.1. General Discussion of the Experiments Relative to the Literature

In this study, two important aspects regarding performance can be discussed: (i) the effect of the type of stimulus used, (ii) the effect of the size of the target stimulus appearance surface and (iii) the effect of the salience—manipulated through the color—of the target stimulus. First, the results obtained in Experiment 1 regarding the type of stimulus employed—with no significant effect on the performance of the system using an ERP-BCI under the SCP (faces versus radar planes)—could be in line with previous work, which also has not found that face stimuli offers significantly superior performance to alternative stimuli [34,35]. Therefore, the use of radar planes as visual stimuli (or those employed by the display at https://www.flightradar24.com, accessed on 10 May 2022) might be appropriate in the use of an ATC managed through an ERP-BCI. Second, based on Experiment 2, the size of the stimulus appearance surface showed a significant effect on performance (E2-RS and E2-YS versus E2-RL). Therefore, the size of the surface to be monitored using an ERP-BCI is a relevant factor that should be considered in future ATC scenarios. Third, regarding the color factor (E2-RS versus E2-YS), it was interesting that there was not a significant effect on the ERP-BCI performance, because we hypothesized that a lower salience of the stimulus to be attended to leads to a decrease. It is possible that, because the area of the E2-RS and E2-YS conditions was relatively small, the use of colors did not pose additional difficulty; this may be corroborated through the missing planes variable, because both conditions produced similar results (0.25 for E2-RS and 0.25 for E2-YS). Therefore, for future proposals, it would be interesting to study in more depth (e.g., more participants and runs) the effect of stimulus salience (e.g., manipulating color) on larger areas.

Considering both experiments together, it is interesting that, in Experiment 1, the stimulus location was not a relevant factor (79% for E1-known and 76% for E1-unknown), but it was in Experiment 2 (around 64.5% for E2-RS, 67.5% for E2-YS and 41% for E2-RL). The context in Experiment 1 was significantly different from that in Experiment 2. In Experiment 1, the conditions presented an interface without distractors, whereas Experiment 2 used an ATC scenario, which could imply additional difficulty. Therefore, according to the results obtained in the present study, the use of an interface overloaded with stimuli could hinder the task, especially when the size of the surface to be surveyed increases (Experiment 2), whereas if the interface is free of distractors—as in the case of a black background—the size of the surface to be surveyed is not a relevant factor (Experiment 1).

Regarding the ERP waveform measured in each experiment, only Experiment 2 showed significant differences in the amplitude difference variable. Specifically, there were significant differences in every channel for the P300 component (around 380–520 ms, depending on the channel), for which the E2-RL condition exhibited a lower amplitude. These results indicate that the grand average of the ERP waveform is negatively affected by the size of the surveillance area. However, we cannot be certain whether the reduced amplitude in the ERP waveform associated with the presentation of target planes in condition E2-RL is due to (i) the fact that planes detected in a larger surveillance area elicit a smaller P300, or (ii) the possibility that this reduced amplitude in the condition is due to averaging the signal of correctly detected target planes with those that were not detected by the user.

Regarding the number of missed planes evaluated in Experiment 2, significant differences in plane perception were found. Although the differences were only found between the E2-YS and E2-RL conditions, it can be affirmed that the size of the monitored area negatively influenced the detection of new planes presented on the map. Due to the fact that only two participants had one error each in the E2-RS condition, only one participant had two errors in the E2-YS condition, and only one participant detected all the planes in the E2-RL condition, it can be inferred that the absence of significant differences between E2-YS and E2-RL may be due to the small sample size. It should be emphasized again that the poor perception of the stimuli in E2-RL may cause the differences in the variables reported above (accuracy and ERP waveform). If the user is not aware of the appearance of the planes, it is unlikely that the ERP-BCI can detect them.

One of the particularities of an ATC (or any application that requires fast detection of a specific stimulus, such as warning or error messages) is that stimuli should be detected with a single presentation (i.e., under single-trial classification). Compared with studies that used single-trial classification [23,25], the results of the present work are lower than expected, especially in Experiment 2. Previous studies that used single-trial classification had an average of around 80%; however, in the present study, the best accuracy was only around 66% (E2-RS and E2-YS, both conditions with a reduced stimulus surface area). Nevertheless, as described in Section 1, the previous studies did not present the characteristics of an ATC scenario, and this difference could explain the decrease in performance. For example, the results of the present work could be explained by factors related to (i) the use of a stimulus-rich background (i.e., the map), (ii) the presence of distractor stimuli while waiting for the appearance of the target stimulus (i.e., other planes moving through the scenario) or (iii) not knowing the exact position where the target stimulus would appear, even if the area of appearance was reduced. Other factors that should be considered for modification—as they have been proved to affect performance—are, for example, the size of the stimulus [18], its color [36] or brightness [20].

### 4.2. Limitations of the Present Study

The present work used an initial and gradual approach to study visual variables in the context of an ERP-BCI used to detect new elements in an ATC scenario. We must admit that this initial approach has some limitations that should be mentioned and discussed.

First, we obtained the display used for the ATC from the web application https://www.flightradar24.com, accessed on 10 May 2022. These displays are not necessarily the most suitable for BCI control nor the same as those that meet the requirements of a real professional interface for managing an ATC. This is an inherent problem when it is desired to transfer findings from one type of display to another, as several visual factors can affect ERP-BCI performance (e.g., variations in background [37], contrast between stimuli [20] or temporal parameters of presentation [38]). Therefore, we advocate for the careful translation of the results among studies.

Second, there are two possible limitations in relation to the studied sample. On the one hand, it should be recognized that, in real scenarios, ATC users are professionals with a high degree of experience in the use of these devices. However, in the present study, the users were not ATC professionals and had varying experience in the use of an ERP-BCI. Therefore, we cannot exclude the possibility that, through extended training in the use of the system, the performance may be better. Expertise is an important factor for the performance of controlling an ATC [39]. Therefore, due to the preliminary and progressive approach of the present study, and the lack of experience by participants using these systems, we advise that future studies attempt to explore in more detail some of the hypotheses stated here.

Third, the classifier that we used is a standard one in an ERP-BCI. After a certain time, the classifier had to select the stimulus that was considered most appropriate as the one desired by the user. Although in some of the conditions the non-target stimuli were not visible (E1-known, E1-unknown, E2-RL, E2-RS and E2-YS), this did not mean that they were absent for BCI2000. Therefore, although the user only perceived the presentation of the desired stimuli (targets), the non-target stimuli were also presented to the classifier (but were invisible to the user). In other words, at the end of the run, the classifier chose the most likely stimulus from among the visible (target) and non-visible (non-target) stimuli. This classification system is different from when the classifier must only discriminate between the detection of a target stimulus versus the selection of nothing. We understand that this presents a problem for running a real ATC application based on an ERP-BCI. However, for the study of visual variables, this approach is not an issue because it is comparing performance by manipulating specific variables to determine if there is a significant effect. In other words, the present work did not focus on absolute performance, but rather on relative performance between different conditions.

## 5. Conclusions

The present work represents the first approach—at the stage of perception of the SA framework—to the implementation of a visual ERP-BCI for the detection of new planes using an ATC. It also shows the importance of the size of the surveillance area in the control of these applications as a crucial variable. The performance shown confirms that this topic is a challenge for the BCI domain under single-trial classification. However, the results are promising enough to continue this research topic. As the combination of an ATC and a BCI is a relatively novel area, there is considerable scope for future proposals. For instance, a proposal could advance to the next level of the hierarchical model of situation awareness, i.e., in the comprehension of the served elements. This second level of the SA framework can be evaluated, for example, using different types of planes to be categorized by the user (e.g., planes or civil drones). Moreover, future work can focus on how to improve the performance of these systems through what has been previously studied in other types of BCI devices (e.g., the spellers, which are the most studied ERP-BCI applications [40]). Some of these improvement proposals include those related to human factors [41] as well as different signal processing and classification techniques [42]. BCI systems have been used previously in the field of ATC. However, they have been used only for the purpose of assessing the cognitive state of users (assessment of mental workload [14] or the presence of microsleep states [15]). Therefore, it would be interesting if future proposals would use a BCI with the dual purpose of (i) measuring the cognitive state of the user and (ii) supporting the correct perception of stimuli at the interface. In short, the use of an ERP-BCI for stimulus detection in an ATC is an interesting area. The present work has shown that (i) the presentation of a new plane in an ATC produces an ERP waveform discriminable by a BCI system and that (ii) the size of the surveillance area is negatively related to the performance of these systems.

## Figures and Tables

**Figure 1 brainsci-13-00886-f001:**
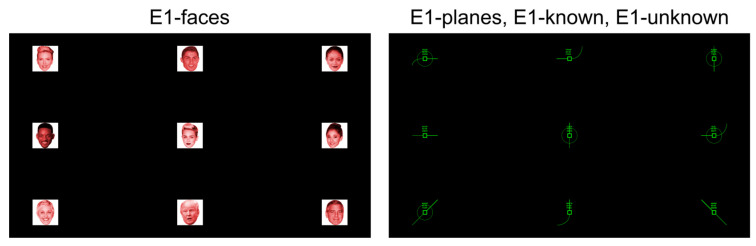
Stimuli and locations used to present them on the screen in Experiment 1. The E1-faces condition used celebrity faces, whereas the E1-planes, E1-known, and E1-unknown conditions used stimuli that simulated those used on flight radar. Images of celebrity faces are pixelated for copyright reasons. The celebrity faces are as follows (from left to right and from top to bottom): Scarlett Johansson, Cristiano Ronaldo, Rihanna, Will Smith, Miley Cyrus, Ariana Grande, Ellen DeGeneres, Donald Trump, and George Clooney.

**Figure 4 brainsci-13-00886-f004:**
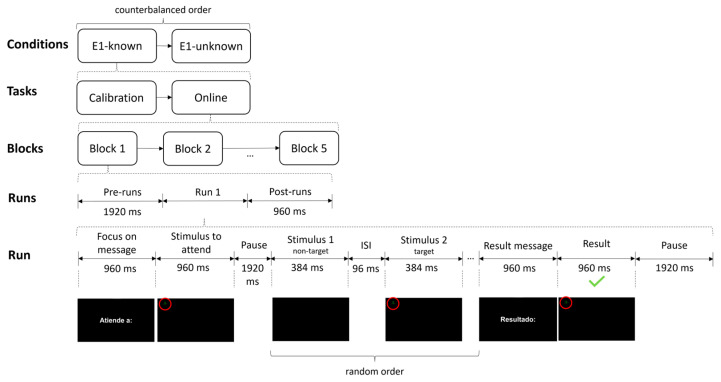
The procedure and timing used in the E1-known and E1-unknown conditions of Experiment 1. Specifically, the figure shows the execution of the first selection of the E1-known condition during the online task. Due to the small size of the stimulus in the figure, compared with when it was presented on the screen during the experiment, the stimulus is marked with a red circle. ISI stands for inter-stimulus interval.

**Figure 5 brainsci-13-00886-f005:**
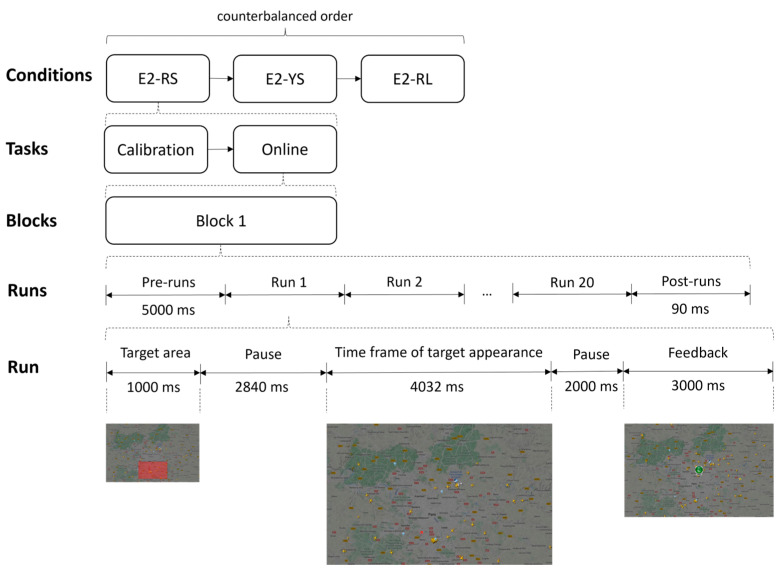
Procedure and timing used in Experiment 2. Specifically, the figure shows the execution of the first selection of the E2-RS condition during the online task. The target stimulus appeared at one of these nine instants (ms) within a “Time frame of target appearance” window: 0, 448, 896, 1344, 1792, 2240, 2688, 3136, 3584. The target stimulus remained on the screen until the block ended or exited at the edge of the screen. ISI stands for inter-stimulus interval.

**Figure 6 brainsci-13-00886-f006:**
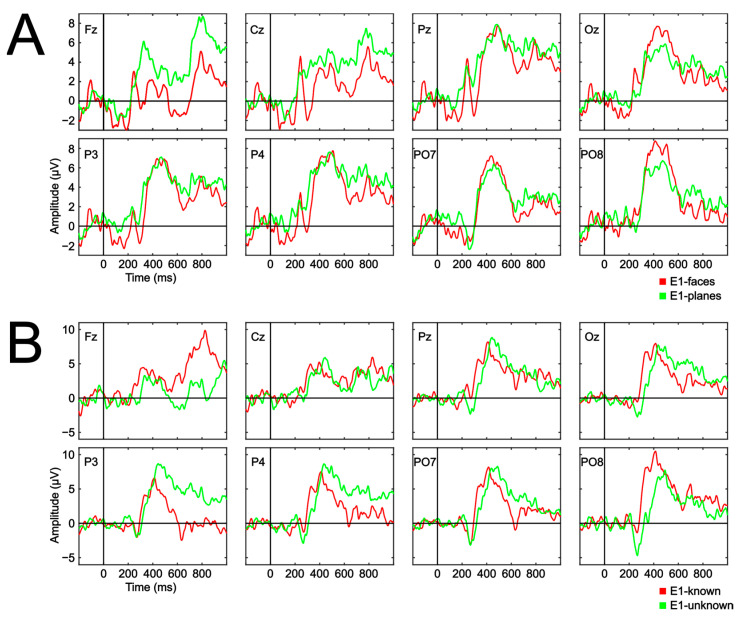
Grand average event-related potential waveforms (μV) for the target ERP waveform amplitude, in all used channels, for the two comparisons carried out in Experiment 1: E1-faces versus E1-planes (**A**) and E1-known versus E1-unknown (**B**). The statistical analyses did not indicate significant differences in any of the time intervals between the target stimuli of the compared conditions in each channel. The false discovery rate (FDR) correction method was applied.

**Figure 7 brainsci-13-00886-f007:**
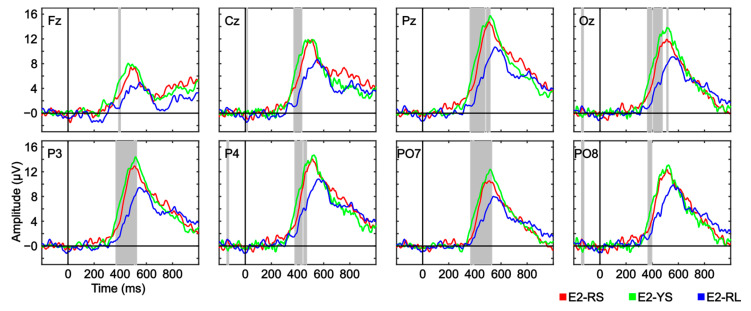
Grand average event-related potential waveform (μV) for the target ERP waveform amplitude, in all used channels, for the three conditions used in Experiment 2: E2-RS, E2-YS and E2-RL. Significant intervals are denoted with a grey background for the relevant time interval. The false discovery rate (FDR) correction method was applied. The results of participant E203 were excluded from the analysis following visual inspection, as their signal exhibited a periodic signal at 25–30 Hz that significantly impacted the grand average.

## Data Availability

The data presented in this study are available on request from the corresponding author.

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
