# Peer review of "Evaluation of Single-Trial Classification to Control a Visual ERP-BCI under a Situation Awareness Scenario"

_brainsci, 2023, doi:10.3390/brainsci13060886_

Round 1

Reviewer 1 Report

1. The introduction is very limited regarding the EEG/ERP content. If the title includes ERP then it is expected to see in the Introduction a more detailed description of ERPs (and not the issues about the need for repeated presentation of the stimulus). Apparently, the authors do not distinguish between exogenous and endogenous ERPs. It could very well be that the exogenous ERP could be a more efficient classifier for novelty?

For example, authors talk about the specific location of a stimulus but they do not review relevant ERP literature in this regard. Furthermore, for stimuli they used faces and motion onset of the airplanes but did not review well know N17o for faces or N200 for motion onset

Methods

Sample size: unacceptable. In this day and age submitting a manuscript with a sample size lower than 10 subjects can not be considered appropriate.

ERP processing:

a. missing

b. it is ridiculous averaging over 5 repeats 

Single-trial classifier: missing details of how the classification was conducted, e.g., single sampling point?

Results

Amplitudes of the ERP waveforms: 

a. lack of any data and statistics for ERPs. It is difficult to believe that there are no significant differences between ERPs for faces and planes as presented in Fig 6 A at Cz and Pz channel.

b. their ERPs are questionable at best. In the Figure 6 A authors show the ERPs for faces and airplanes but the waveforms look strange, not real ERPs. For example, there are missing expected exogenous components for airplanes

c. missing baseline correction for ERP waveforms presented in the Figure 6.

d. The ERP figures are not properly presented -missing pre-stim period. 

In sum, results are questionable at best. Small sample size, inappropriate ERP analyses and missing data and statistics should be corrected

Discussion 

Unacceptable lack of discussion related to ERPs. There is excessive talk about the flight control issues but a single statement about ERP, more specifically about P300. The author wrote: " It is possible that by decreasing the salience of the target stimulus—and thus increasing the difficulty of the task—the user actively invests more attentional resources, which affects the amplitude of the P300 component [39]." In this sentence, they are referencing a publication from 1988?? It has been well established that the exogenous P1, and N1 components are much more sensitive to attentional manipulations. Authors need a significant increase in their knowledge related to EEG/ERPs.

Author Response

Please, find attached the document "responses_brainsci-2367657-Reviewers.pdf".

Reviewer 2 Report

The manuscript can be considered interesting for the Brain Science readers. However, it needs significant improvements. 

Please, find my detailed consideration below.

  1. To make it easier for future readers, the primary motivation for this study needs to be made clear in the introduction section. They can then discover the main idea and the method used to resolve the issue at hand. Try to present your contribution in a challenging and helpful way, highlighting the issue with the prior studies and your innovative solution.
  2. In my opinion, the introduction should be further improved to better specify the work's motivation and the methodology's characteristics. As it is, it is not clear how it works. Additionally, the author should state clearly in which aspect this work extends state of the art, i.e., highlight the novelty.
  3. The methodology, novelty, and experimental findings for each paper should be explicit in the literature review. Highlight more clearly in a few lines what general technical shortcomings in previous works were found to have prompted the development of the suggested approach at the conclusion of related works.
  4. The experimental section needs some clarifications. Further details should be provided. The analysis and explanation of the test results need to be supplemented.
  5. What is your fitness function? what is the outcome of the proposed algorithm?
  6. What is your technical experimental setup? give full details.
  7. In order to support the suggested work, the results section should be connected to a discussion section.

English language needs only modest correction.

Author Response

(The authors gave the same response as above.)

Round 2

Reviewer 1 Report

The authors appropriately responded to this reviewer's comments. In particular, they increased the sample size and improve ERP data processing.